# BuildingBRep-11K: Precise Multi-Storey B-Rep Building Solids with Rich Layout Metadata

**Yu Guo**[1,2]  **Hongji Fang**[1]  **Tianyu Fang**[1]  **Zhe Cui**[1]

[1] College of Architecture and Urban Planning, Tongji University
[2] College of Design and Engineering, National University of Singapore

{waterice, fanghongji, 2410319, cuizhe}@tongji.edu.cn

## Abstract

With the rise of artificial intelligence, the automatic generation of building-scale 3-D objects has become an active research topic, yet training such models still demands large, clean and richly annotated datasets. We introduce BuildingBRep-11K, a collection of 11 978 multi-storey (2-10 floors) buildings (about 10 GB) produced by a shape-grammar-driven pipeline that encodes established building-design principles. Every sample consists of a geometrically exact B-rep solid-covering floors, walls, slabs and rule-based openings-together with a fast-loading .npy metadata file that records detailed per-floor parameters. The generator incorporates constraints on spatial scale, daylight optimisation and interior layout, and the resulting objects pass multi-stage filters that remove Boolean failures, undersized rooms and extreme aspect ratios, ensuring compliance with architectural standards. To verify the datasets learnability we trained two lightweight PointNet baselines. **(i) Multi-attribute regression.** A single encoder predicts storey count, total rooms, per-storey vector and mean room area from a 4000-point cloud. On 100 unseen buildings it attains 0.37-storey MAE (87 % within $\pm1$), 5.7-room MAE, and 3.2 m$^2$ MAE on mean area. **(ii) Defect detection.** With the same backbone we classify GOOD versus DEFECT; on a balanced 100-model set the network reaches 54 % accuracy, recalling 82 % of true defects at 53 % precision (41 TP, 9 FN, 37 FP, 13 TN). These pilots show that BuildingBRep-11K is learnable yet non-trivial for both geometric regression and topological quality assessment.

## 1 Introduction

Artificial intelligence is revolutionizing various industries, and its role in assisting architectural design has become a significant focus in architectural research. In this context, the term "architecture" specifically refers to the discipline of designing and shaping physical structures and spaces, encompassing creative, functional, and spatial considerations such as aesthetics, human interaction, environmental integration, and cultural expression. To avoid confusion with the term "architecture" used in computer science (e.g., system or network architecture), this article will adopt the word "building" to represent the architectural domain. However, "building" retains the essential attributes of architecture, including its tangible forms, spatial relationships, and design intentionality, ensuring alignment with the original conceptual depth of the field.

**Existing building dataset** Datasets form a fundamental pillar of deep learning-based building generation tasks. Currently, existing 2D building datasets such as RPLAN [1], LIFULL[2], CubiCasa5K[3], MSD[4], and P-PLAN[5] are primarily derived from real-world residential or public building floor plans, offering a high degree of spatial plausibility. However, collecting, cleaning, and standardizing real-world datasets present significant challenges and are often labor-intensive and

time-consuming. In contrast, synthetic datasets offer advantages in consistency and normalization. High-quality synthetic datasets, such as ALPLAN[6], can also achieve strong spatial rationality and interpretability.

In recent years, several large-scale building and city-level 3D datasets have been introduced, including BuildingNet[7], Building3D[8], RealCity3D[9], City3D[10], and UrbanScene3D[11]. These datasets mainly encompass mesh, point cloud, and voxel formats, and have been widely adopted for tasks in computer vision and graphics, such as semantic segmentation, scene understanding, and model generation. Boundary representation (B-rep) is a three-dimensional modeling format for describing geometric shapes, with broad applications in Computer-Aided Design (CAD). However, acquiring such precise models from the real world is extremely challenging, resulting in a scarcity of 3D building datasets with B-rep representations. This limitation has, to some extent, hindered the advancement of deep learning research within this domain.

**Existing deep-learning algorithm**  Another crucial aspect of deep generative tasks is the generation model itself. In the context of 2D floor plan generation, many well-established generative models, such as GANs [12–15], GNNs[16, 17], and diffusion models[18, 19], have been widely adopted. These models can produce high-quality floor plans with reasonable layouts and complete functional zones, and have been extensively explored in architectural design. However, for 3D building model generation, while methods such as Score Distillation Sampling (SDS)[20–22] can produce high-fidelity models, they often fall short in geometric precision. Algorithms capable of handling precise formats like B-rep are still rare[23], and further research attention is needed to advance this area.

## 2   Data generation pipeline

The samples we aim to generate must adhere to the scale of building design, resembling standard BIM models that encompass both internal spatial enclosures (walls and slabs) and functional openings (doors and window apertures) at appropriate locations. To achieve this, we first established general building design requirements and simulated a typical design workflow: single-floor planar framework - wall and opening layout - multi-tiered vertical composition. The planar framework integrates residential design scales, while the door-and-window opening incorporates fundamental daylighting and spatial usability criteria. Multi-tiered vertical composition adopts a tiered setbacks strategy to maximize open spaces across floors. These rules are combined with randomized parameters to form a shape grammar-based generative system. Using a dual parametric-and-random drive, the system can synthesize an unbounded number of samples while allowing any specific instance to be reproduced exactly via its random seed.

### 2.1   Building design principle

The design guidelines follow typical residential dimensions, which are relatively flexible and easily adapted to other building types. To establish a consistent reference condition, we assume every building sits just north of the Tropic of Cancerthereby ensuring the suns path always lies to the south. Under this standardized solar geometry, providing south-facing windows becomes critically important simply to admit direct sunlight, independent of any specific climate considerations. The generation of each sample is shown in Figure 1.

For each floor plan, the layout skeleton is grown iteratively from an original, fixed rectangle that represents the core tube. The detail of this step will be illustrated in Section 2.2.1. This method resonates strongly with the growing house philosophy, which conceives of a building as an aggregation of discrete, independent modulesan approach ideally suited to generating a large number of diverse samples.

Each floor is then extruded into a solid B-Rep, with window openings cut into exterior walls and door openings in interior partitions. The full procedure is detailed in Section 2.2.2. Our guiding principle is to guarantee that every room remains accessible, receives sufficient daylight, and avoids redundant openings that could complicate interior finishing.

Finally, the individual floor solids are stacked to form the complete building mass. Following our tiered-setback strategy, floors with more rooms are placed beneath those with fewer rooms, creating

open terraces at each setback level. Additionally, on every storey the slab of the coretube rectangle is opened to provide a vertical atrium. On the ground floor, our program selects an exterior wall segment-based on the outer footprint-to create a main entrance, ensuring that the first level remains connected to the outside. The detailed rules governing these operations appear in Section 2.2.3.

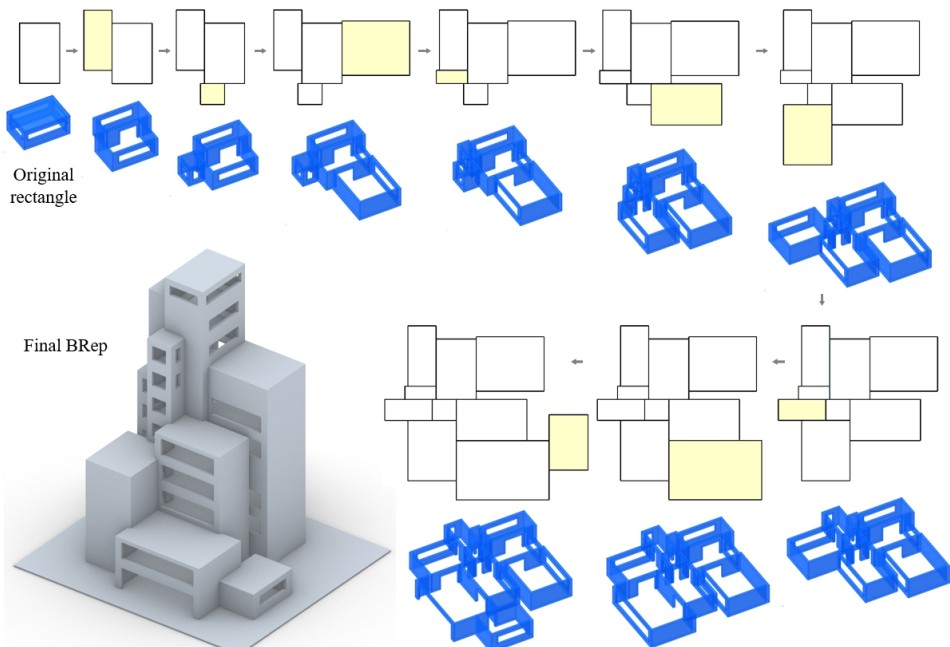

Figure 1: The production of a building BRep solid

## 2.2 Details in generator

The data generator is organized into four sequential modules. The first module constructs the planar skeleton; the second converts this skeleton into solid walls and inserts door and window openings; the third stacks the resulting storey solids to synthesize the complete building B-Rep; and the fourth performs post-processing, exporting the final B-Rep together with its metadata while filtering out any instances that cause conflicts within the brep solids. Unless otherwise specified, all B-Rep dimensions reported in this paper are given in metric units (metres).

### 2.2.1 Plan skeleton

We implement our floorplan skeleton as a planar shape grammar driven by two production rules-"concaveangle expansion" (yin rule) and convexangle expansion (yang rule)-each of which grafts exactly one new rectangular room onto the existing polyline footprint per iteration. In code, we begin with a single original rectangle (the core tube) and then, at each step, classify a randomly chosen vertex as concave or convex, apply the corresponding expansion rule, clean up any redundant vertices, and repeat until a suitable plan size is reached.

**Vertex classification.** We examine each polyline vertex and classify it as concave (yin) or convex (yang) by testing whether a tiny neighborhood around the point lies entirely on the planar surface, as shown in Figure 2.

**Rule application** **For concave expansion:** at a concave vertex $v$, we measure the two adjacent edge lengths to decide how large a new room to graft; we construct that room as a parallelogram by shooting out from $v$ along the bisected angle, then union it with the existing footprint. **For convex expansion:** at a convex vertex we similarly choose an anchor point (either the midpoint of the long edge or the adjacent vertex) and project outward to form a new parallelogram, again unioning it into the skeleton.

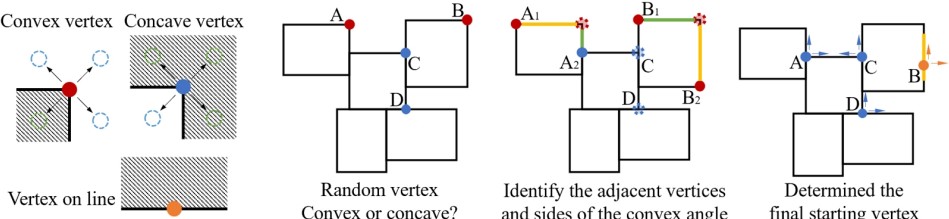

Figure 2: Define vertex type and corresponding rectangle generation

**Normalization and cleanup**  After each union we remove redundant vertices (collinear or coincident points) and, if tiny notches remain, invoke a smallsegment filler to snap them closed.

Because each iteration replaces one footprint polyline *P* with a larger polyline *P'* via one of two rewriting rules, this process exactly matches the abstract form of a contextfree shape grammar:

$$P \longrightarrow P \cup R_{\mathrm{yin}}(v) \quad \text{or} \quad P \longrightarrow P \cup R_{\mathrm{yang}}(v). \tag{1}$$

where $R_{yin}$ and $R_{yang}$ are the two parallelogramaddition productions anchored at vertex *v*. A fixed random seed ensures reproducibility of the sequence of rule applications.

Although the shape grammar permits unbounded growth, in consideration of realistic architectural proportions and computational performance constraints, we impose an upper limit of ten rectangle-shalting the expansion once ten modules have been generated.

### 2.2.2  Level solid

In our pipeline, the polyline skeleton is offset equally on both sides and then lofted and extruded to form solid walls. Next, door openings are carved into interior walls to guarantee every room remains accessible without interrupting the circulation flow. Finally, each exterior wall segment undergoes a two-stage process-window generation followed by window pruning-as detailed below. Figure 3 shows an example of the opening method.

**Window Generation**  For each segment of the external wall, we measure its length and orientation as wall parameters, then classify it into one of the four cardinal orientations-north, east, south, or west. These four orientations are then organized into two groups: east-west and north-south, each sharing its own set of window parameters and associated thresholds. Our window parameters include the sill height (bottom elevation of the opening), the wallsetback inset distance (how far the window is offset from each end), and the overall window height. By assigning distinct threshold values to the eastwest group versus the northsouth group, we compute the appropriate parameter values for each exterior wall segment. Wall lengths in each orientation group are partitioned into three bins based on predefined thresholds, each bin corresponding to a standard window type whose exact dimensions can be tweaked in the generator. In our current configuration, north-south-facing windows are generally larger than their east-west counterparts to reduce western solar gain; most windows are centered along the wall span, whereas the smaller eastwest windows are deliberately offset toward the southern end.

**Window Pruning**  To avoid over-fenestration (which impedes furniture layouts), we then iterate over each rooms set of candidate windows and apply a set of rules. First, we examine each rooms set of candidate windows and apply a hierarchical sizebased filter: if a room has two to four openings, we first identify the widest aperture; when its span is at least 3, all smaller openings are discarded; if the maximum span lies between 1 and 3, only the largest and smallest openings are retained; and if every opening is under 1, thenprovided there are more than twoonly those on the northsouth façades remain, otherwise all windows are kept.

### 2.2.3  Create BRep

The storeys are assembled according to the tiered-setback principle, with the core tube kept vertically aligned throughout. The ground-floor slab is generated parametrically by offsetting the initial bound-

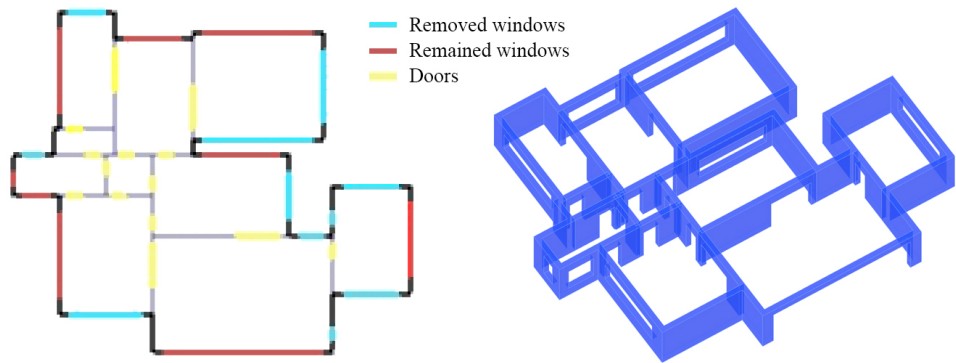

Figure 3: Example of door opening, initial and removed window opening

ing box outward by 3 modular units, thereby producing an enlarged base surface that represents the ground plane. Entrance placement follows an algorithmic protocol: exterior walls exceeding four units in length are prioritized, and the segment nearest the geometric centroid is selected for portal insertion, ensuring both symbolic visibility through its axial positioning and functional efficiency via optimized spatial connectivity.

The hybrid stochastic-parametric generation framework incorporates real-time collision detection, automatically suspending horizontal expansion upon identifying topological conflicts in the planar skeleton. As illustrated in Figure 4, a conflict occurs when generating the sixth rectangle, which triggers system intervention and results in a five-storey built form with setback-adjusted massing. The growth termination at this conflict threshold preserves tectonic integrity by maintaining load-path continuity, programmatic adjacencies, and proportional harmony, embodying an adaptive architecture that reconciles generative exploration with constructional logic through embedded self-regulating mechanisms.

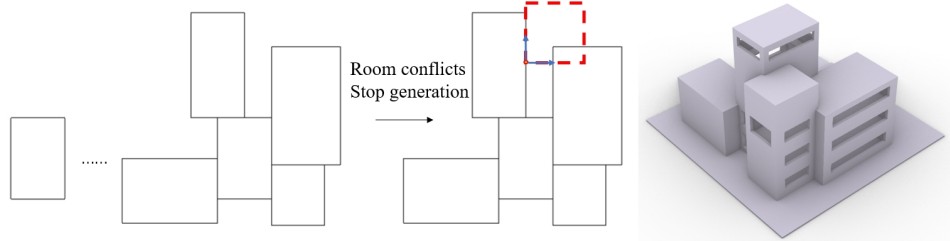

Figure 4: Example of a 5-storey building under the topological conflicts while generating the 6th rectangle.

### 2.2.4 Post-processing

After the storey solids are merged, redundant edges are removed and all gaps are sealed to obtain a watertight, geometrically exact building solid. Each solid is then exported as a storage-efficient B-Rep file, while its metadata-including total storey count, per-storey rectangle dimensions, and door-window specifications-are written to an accompanying Excel sheet. Using these metadata, we automatically discard any instance that contains rooms with out-of-range dimensions. The present study releases a curated corpus of 11 978 such B-Rep models (about 10 GB in total), hereinafter referred to as BdBR-11K. The parameters of every case are further consolidated into a single META file to facilitate downstream querying and analysis.

## 3 Stadistical Analysis

The BdBR-11K dataset, generated through stochastic parameter configuration, exhibits substantial typological diversity in building stratification, spatial metrics, and footprint configurations. This

Table 1: Storey distribution of BdBR-11K

| Storey | 2 | 3 | 4 | 5 | 6 | 7 | 8 | 9 | 10 | Total |
|---|---|---|---|---|---|---|---|---|---|---|
| Sample number | 88 | 600 | 929 | 1158 | 1205 | 1047 | 861 | 705 | 5385 | 11978 |

curated collection demonstrates geometric precision with richly annotated architectural parameters while maintaining large-scale volumetric characteristics, rendering it particularly suitable for training deep learning networks in architectural computation and generative design applications. The hierarchical metadata structure enables systematic extraction of parameter-specific samples (e.g., floor configurations, fenestration ratios, spatial typologies), thereby enhancing training controllability through targeted feature-space manipulation.

**Storey distribution**   Table 1 shows a right-skewed storey distribution: the dataset contains 11 978 buildings, with low-rise (2-3 storeys) making up only 688 samples (less than 6 per-cent), mid-rise (48 storeys) 5 200 samples (about 43 per-cent), and 10-storey towers dominating at 5 385 samples (about 45 per-cent). Thus BdBR-11K is biased toward taller forms, providing abundant multi-storey examples while still retaining a spectrum of lower heights for diversity. Because buildings of different heights occur with unequal frequencies in practice, the resulting skew is both expected and reflective of real-world distributions.

**Room area distribution**   The room-area density curve (Figure 5a) shows a pronounced peak around 15 $m^2$, indicating that the vast majority of rooms in BdBR-11K cluster near the floor-space commonly assigned to compact bedrooms or studies. Density then tapers gradually between 20 $m^2$ and 40 $m^2$, covering typical living-room sizes-before approaching zero beyond 80 $m^2$. The unimodal, left-skewed profile implies the dataset emphasises realistic, space-efficient residential units while still providing a usable tail of larger, specialty rooms.

**Footprint distribution**   The storey (floor)-area density curve (Figure 5b) exhibits its maximum at roughly 270 $m^2$, with a steep rise from the minimum recorded values (about 50 $m^2$) and a long, gently declining tail that extends past 450 $m^2$. This right-skewed distribution suggests that most generated floor plates correspond to mid-scale apartment or office levels, whereas substantially larger footprintsappropriate to podiums or communal facilities-occur less frequently but remain available for model training that requires broader volumetric variation.

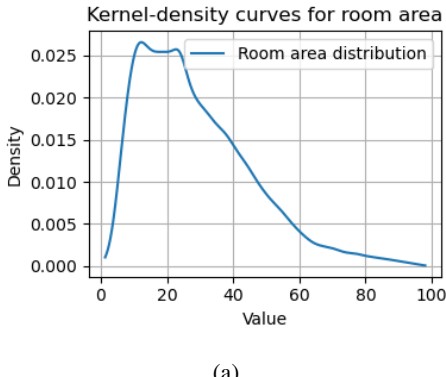
(a)

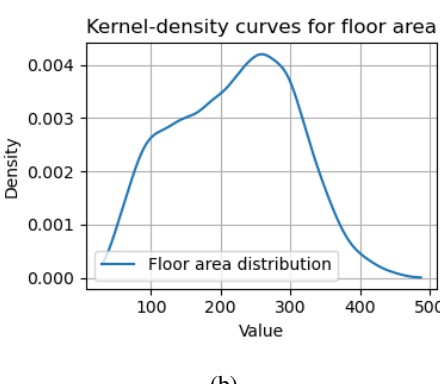
(b)

Figure 5: (a) The room-area distribution; (b) The floor area distribution.

# 4   Baseline Experiments

To demonstrate the suitability of our dataset for learning tasks, we implemented two proof-of-concept baselines using a PointNet architecture.

### 4.1 Predicting four geometric attributes from a single B-Rep Problem definition.

For every building B-Rep we first sample 4 000 surface points, normalise them to the unit cube, and feed the resulting point cloud to a *PointNet-Multi* network. The backbone is the standard PointNet encoder (spatial T-Net, three $1 \times 1$ convolutions, global max-pool) that produces a 1 024-dimensional feature vector. Four lightweight fully connected heads are attached; their targets and loss functions are summarised in Table 2. The four losses are summed with equal weight, which empirically balances gradient magnitudes without tuning. Training uses Adam (learning rate $10^{-3}$, weight decay $10^{-4}$), batch size 32, and an 80:10:10 train / validation / test split; longer schedules can further improve accuracy, but a 20-epoch run already recovers both discrete and continuous attributes directly from raw geometry. These results confirm that BdBR-11k supports multi-objective 3-D deep-learning tasks without any hand-crafted features.

Table 2: PointNet-Multi prediction heads and corresponding loss terms.

| Head name | Predicted quantity | Loss function |
|---|---|---|
| storey-cls | storey count (2-10 floors) | cross-entropy |
| room-tot | total room number | $L_1$ error |
| room-per | 9-dim vector of per-storey room counts | mean-squared error |
| avg-area | mean room area ($\mathrm{m}^2$) | mean-squared error |

The four losses are summed with equal weight, which empirically balances gradient magnitudes without manual tuning. We optimise with Adam (learning rate $10^{-3}$, weight decay $10^{-4}$), using a batch size of 32 and an 80:10:10 train/validation/test split; the quick proof-of-concept runs for 20 epochs, though longer schedules further improve accuracy. We report overall storey-classification accuracy, RMSE on total-room count, and MAE on mean room area, all on the held-out test fold. Despite its compactness, the model successfully recovers both discrete and continuous architectural attributes directly from raw geometry, demonstrating that BdBR-11K is suitable for multi-objective 3-D deep-learning tasks without hand-crafted features.

**Evaluation protocol** After training had converged we drew a random test batch of 100 buildings from the full 11 978 sample corpus. The overall evaluation results are summarised in Table 3.

Table 3: Overall test performance on the 100-building hold-out set

| Metric | Value | Comment |
|---|---|---|
| Storey accuracy | 0.64 | exact-match fraction |
| Mean absolute error (storey) | 0.42 | floors |
| Root-mean-square error (room total) | 10.2 | rooms |
| Mean absolute error (avg. room area) | 4.0 | $\mathrm{m}^2$ |
| Mean absolute error (per-floor room count) | 0.33 | rooms per floor |

**Metric computation** The four error metrics in Table 3 were obtained from the spread-sheet according to

$$\text{storey\_error} = |\text{pred\_storey} - \text{real\_storey}|, \tag{2}$$

$$\text{roomtot\_error} = \text{pred\_room\_tot} - \text{real\_room\_total}, \tag{3}$$

$$\text{avgarea\_error} = \text{pred\_avg\_area} - \text{real\_average\_area}, \tag{4}$$

$$\text{per-floor MAE} = \frac{1}{10} \sum_{i=1}^{10} |\text{pred\_room\_per\_i} - \text{real\_room\_per\_i}|. \tag{5}$$

Because the ground-truth per-floor columns are empty in the sheet, the *real* per-floor room counts were reconstructed from the ground-truth storey count $s$ by the deterministic pattern $(s, \ s - 1, \ \ldots, \ 1, \ 0, \ 0, \ \ldots)$.

**Discussion**  On the 100-building hold-out set the baseline PointNet-Multi achieves **height classification accuracy of 87 % within $\leq 1$ storey** and keeps absolute errors on room totals and average areas within a range that remains usable for downstream tasks such as early volumetric code checking or preliminary cost estimation.

## 4.2  Detecting defects

The second baseline targets a *binary quality-control task*: given a building B-Rep, decide whether the model is **GOOD** (topologically watertight and fully enclosed) or **DEFECT** (at least one exterior face is missing, yielding an open shell). The motivation is practical automated BIM pipelines and downstream simulation tools require clean, manifold solids; a fast pre-screening stage can filter out corrupted uploads before expensive meshing or analysis is attempted. We cast the problem as 3-D shape classification: each input B-Rep is uniformly rasterised to a $4,000$-point surface cloud, normalised to the unit sphere, and fed to a lightweight PointNet encoder (*shared MLP $\rightarrow$ max-pool $\rightarrow$ FC$_{128}$*) followed by a two-way soft-max head. The network is trained from scratch on the BdBR-11K training split with a cross-entropy loss; data augmentation comprises random rotation about the $z$-axis and Gaussian jitter.

**Defect vs. good baseline (100-sample test).**  The `test_result.csv` file contains `filename` and predicted `prediction` labels. A sample is considered *DEFECT* if `filename` contains the substring "_def", otherwise it is *GOOD*. On the 100 random test cases we obtain the confusion matrix

|              | pred DEFECT | pred GOOD |
|--------------|:-----------:|:---------:|
| truth DEFECT | 41          | 9         |
| truth GOOD   | 37          | 13        |

which gives an accuracy of $54\%$, precision $52.6\%$, recall $82.0\%$, and $F_1 = 0.64$.

**Discussion.** The PointNet classifier shows a clear bias toward the *DEFECT* class: it recovers the vast majority of real defects (high recall) but also flags many sound models as defective (low precision). Two factors explain this behaviour.

1. *Geometric signal sparsity.* PointNet receives a uniform 4 k-point cloud sampled from the B-Rep surface. Missing faces typically remove only a few hundred points, so the global point distribution changes little; conversely, open edges introduce noisy point density that the network may interpret as a defect. Thus the network errs on the side of "defect" whenever the surface sampling is ambiguous.

2. *Class-imbalance during training.* In the training split defects out-number good models by roughly 2:1, so the cross-entropy optimum is skewed toward the positive class. A simple re-weighting of the loss or a focal loss would already shift the classifier toward a more balanced operating point.

Even with these limitations, the baseline still achieves an $F_1$ above 0.6-evidence that the dataset does contain exploitable 3-D cues. Future work can improve the result by (i) increasing the point sample near sharp edges, (ii) adding surface normals or curvature as extra channels, and (iii) fine-tuning the decision threshold or using cost-based sampling to restore precision without sacrificing recall.

## 5  Conclusion

We have presented BuildingBRep-11K, the first public corpus of more than eleven thousand watertight, millimetre-accurate B-Rep solids whose geometry is *natively* three-dimensional and accompanied by exhaustive, machine-readable metadata (storey count, room statistics, opening parameters, *etc.*). In contrast to prior floor-plan or mesh resources, every instance is defined by analytic surfaces, trimmed faces and exact topology, making it directly consumable by CAD kernels, physics solvers and emerging B-Rep neural networks. Because the generator exposes all design variables, researchers can sample controlled curricula, inject domain shifts or synthesise unlimited data for self-supervised pre-training. Our lightweight PointNet baselines—multi-attribute regression and defect classification—illustrate only two of many possible benchmark tasks; the same assets naturally

support surface segmentation, edge labeling, Boolean error detection, structural simulation or text-to-shape retrieval. By filling the long-standing vacancy of a large-scale, parameter-rich B-Rep dataset, BuildingBRep-11K lays a common foundation for future work on precise geometric learning.

The current generator encodes the residential massing rules, tiered-setback envelopes and window heuristics. Although this yields diverse yet realistic samples, it under-represents curved facades, non-orthogonal cores and mixed-use programs. Extending the grammar with additional typologies, façade systems, mechanical shafts or stochastic material properties would further enlarge the design space and increase task difficulty. Moreover, coupling the generator with differentiable simulation (daylight, energy, structure) could produce *performance-labelled* data for multi-objective optimisation studies. We will release the full pipeline, hoping the community will refine, remix and augment it; we envision a family of open generators that continuously grow in fidelity and scope, ultimately bridging architectural design practice and data-driven 3-D intelligence.

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

```python
import rhinoscriptsyntax as rs
import random
import System.Drawing as sd
random.seed(randomseed)

def if_yin(cr, pt):

def three_point_parallelogram(pt_m, pt_s, pt_e):

def yin_gen(pl,i):

def yang_gen(pl,i):

def remove_redundant_plpoint(x):

def main_buzu(pl, n):

def fill_yin_once(cr):

def check_yin(cr):

def auto_fill_triyin(cur):

rectangles=[pl]
colors=[sd.Color.FromArgb(255,255,255)]
vs=rs.PolylineVertices(pl)
ps=(vs[0]+vs[2])/2
bu=[]
positions=[ps]
txt=[0]
npls=[]
ver=[]

n=1
#totle=10
cl_x=255
cl_step=int(cl_x/totle)
while n <totle: #default 10
```

(a)

```python
import rhinoscriptsyntax as rs
import scriptcontext as sc
if bool==True:
    win_len=len(window_line)
    rec_len=len(recs)
    i=0
    remove_list=[]
    while i < rec_len:
        pl=recs[i]
        rg=rs.AddPlanarSrf(pl)
        temppt=rs.SurfaceAreaCentroid(rg)[0][2]
        lt={}
        j=0
        while j < win_len:
            l=window_line[j]
            pt=rs.CurveMidPoint(l)
            pt1=rs.AddPoint(pt)
            rs.MoveObject(pt1, rs.AddPoint(0,0,temppt-rs.PointCoordinates(pt1)[2]))
            tem_if=rs.IsPointOnSurface(rg, pt1)
            if tem_if==True:
                winline_len=rs.CurveLength(l)
                pt1=rs.PointAdd(rs.AddPoint(0.1,0,0),pt)
                pt2=rs.PointAdd(rs.AddPoint(0,0.1,0),pt)
                pt3=rs.PointAdd(rs.AddPoint(-0.1,0,0),pt)
                pt4=rs.PointAdd(rs.AddPoint(0,-0.1,0),pt)
                if rs.IsPointOnSurface(rg,pt1)==False: orien='E'
                if rs.IsPointOnSurface(rg,pt2)==False: orien='N'
                if rs.IsPointOnSurface(rg,pt3)==False: orien='W'
                if rs.IsPointOnSurface(rg,pt4)==False: orien='S'
                lt[j]=[orien,winline_len]
            j+=1
        r_c_l=[]
        if len(lt)>=2:
            templenth=0
            for w in lt:
                if lt[w][1]>templenth:
                    longestid=w
                    templenth=lt[w][1]
            if templenth>=3:
```

(b)

Figure 6: (a) Functions in the skeleton module; (b) Part of the remove window code.

## Description of dataset

**Data generator**

The dataset is created based on python-grasshopper of Rhino, which will soon be packed and shared on github. Figure 6 are some of the detailed functions of data generator.

**Code and dataset**

The data can be found at

> https://huggingface.co/datasets/WATERICECREAM/BuildingBRep11k

which contains the whole dataset and part of which used for training and testing. The root directory contains all source files of the dataset: BuildingBRep11k.tar, meta.json, and meta.npy. Furthermore, the task.tar archive includes the dataset specifically designated for Task 2 testing.

The code can be found at

> https://github.com/watericecream/Tasks-of-BuildingBRep11k-dataset

which contains codes of the two baseline tasks, and the needed packages.

