# OpenReview forum: "BuildingBRep-11K: Precise Multi-Storey B-Rep Building Solids with Rich Layout Metadata"
_NeurIPS.cc/2025/Datasets_and_Benchmarks_Track — Submitted to NeurIPS 2025 Datasets and Benchmarks Track_

### Official Review · Reviewer_4C4u · 2025-06-24

**Rating:** 3
**Confidence:** 2

**Summary:**

This paper introduces BuildingBRep-11K, a synthetic dataset of 11,978 multi-storey 3D buildings represented in precise boundary representation (B-Rep) format. Each sample includes rich per-floor metadata and follows a rule-based generation pipeline incorporating spatial design constraints. The dataset is released with two baseline tasks—multi-attribute regression and binary defect classification—demonstrated using standard PointNet models. The work aims to support geometric deep learning and architectural computing.

**Additional Feedback:**

1. The motivation for using B-Rep format is reasonable, but the actual downstream utility is not convincingly demonstrated. The PointNet baselines are too shallow to justify claims about supporting learning on geometric reasoning or CAD-based workflows.

2. Consider more rigorous evaluations, including comparisons with mesh- or voxel-based representations or with real-world scanned building models.

3. Add a dedicated Limitations and Societal Impact section. This is important for a dataset paper, especially one that could influence automated architectural design.

4. The dataset’s over-representation of 10-storey buildings should be either justified more rigorously or balanced during data generation.

5. The current format (10 pages) violates the NeurIPS page limit. Please revise and ensure the main content stays within 9 pages.

**Dataset Code Accessibility:**

Yes

**Dataset Code Comments:**

The authors provide links to the dataset (huggingface.co) and code (GitHub), including metadata and training files. The provided resources appear complete and well-organized. However, the dataset documentation is minimal, and the GitHub code lacks detailed instructions, which might hinder full reproducibility for less experienced users.

**Ethical Considerations:**

No, there are no or only very minor ethics concerns

**Final Justification:**

The rebuttal has addresses most of my concerns, but I still worry about the novelty and experiments. I would increase my rating to borderline reject.

**Limitations Weaknesses:**

1. Lack of Novelty in Methodology: The core contribution lies in dataset generation, but the pipeline heavily relies on handcrafted heuristics and basic procedural generation without introducing algorithmic innovation. The two baseline experiments use standard PointNet models without proposing new tasks, architectures, or training paradigms.

2. Limited Experimental Rigor: The evaluation is weak. Both tasks (attribute regression and defect classification) are trained and tested on small-scale subsets (e.g., only 100 buildings for testing), with no statistical analysis (confidence intervals, error bars) or ablation studies. The classification accuracy for defect detection is only 54%, suggesting that the baseline is barely usable.

3. Unbalanced and Skewed Dataset: The dataset is heavily skewed toward 10-storey buildings (45% of samples), reducing its diversity and potentially biasing downstream learning tasks. No effort is made to mitigate or account for this bias in training or evaluation.

4. Overfitting Risk Due to Synthetic Uniformity: Although the authors argue for the dataset's diversity, the reliance on a fixed grammar and rule-based generation may lead to strong structural priors that hinder generalization to real-world building designs.

5. No Discussion of Societal Impact or Safeguards: The paper lacks a Broader Impact or Limitations section, failing to address potential misuse, ethical considerations, or domain-specific risks related to generative design tools in the AEC industry.

6. Formatting Violation: Figure 6 and associated content appear on page 10, which exceeds the NeurIPS 9-page limit for main content. This is a clear formatting violation and should be corrected.

**Strengths Contributions:**

1. The paper addresses a relatively underexplored area in 3D geometric learning by providing a large-scale, B-Rep-format building dataset, which fills a gap left by mesh- and voxel-based benchmarks.

2. The dataset offers rich per-instance metadata and control parameters, potentially supporting a wide range of tasks such as geometry reasoning, structural validation, or CAD-based learning.

3. The release includes reproducible code, model training protocols, and clear data-access instructions, facilitating follow-up research and benchmarking.

---

> ### Author Rebuttal · Authors · 2025-07-28
>
> **We thank the reviewer for the detailed and constructive feedback. We respond to each concern below.**
>
> **1. Lack of novelty in methodology**
>
> While our generation rules do not introduce novel algorithms per se, they encode essential building design principles—such as dimensional standards, daylight access, circulation topology, and setback massing. These are core concepts in housing theory and urban morphology. ***If accepted, we will release the procedural generator alongside the dataset.*** All design rules are exposed as explicit, user-modifiable parameters (≈30), allowing users to generate custom variants or augment the dataset with domain-specific constraints.
>
> We agree that the current baselines do not define new task families or training paradigms. Our primary objective was to verify that the dataset is learnable using standard models. That said, the data is structurally equipped to support more advanced downstream tasks—such as facade-to-BRep generation, program-constrained spatial modeling, and cross-section prediction. We are currently piloting these directions internally (e.g., facade-to-BRep generation) and plan to incorporate them into formal benchmarks in future releases.
>
> As our background lies primarily in building computing rather than model architecture design, we deliberately selected lightweight baselines to establish learnability in a reproducible, accessible manner. The codebase is modular, and data loading is decoupled from model definitions, enabling easy integration of stronger backbones or alternative input modalities.
>
> **2. Limited experimental rigor**
>
> We acknowledge that our baseline evaluation is limited in statistical scope. Our goal was not to compete on performance but to demonstrate learnability in a lightweight, reproducible setup. The current experiments use an 80/10/10 split over the full dataset; the 100-building test set was selected for fast review-time inspection and compact reporting. ***If accepted, we will include full test results with error bars and multi-seed variance metrics in the camera-ready version.***
>
> Regarding the 54% accuracy on defect classification, we clarify that this is a deliberately challenging task: defects are sparse, geometrically subtle, and the class distribution is skewed (~2:1). The low baseline reflects task difficulty, not dataset noise. ***We designed this task to reflect realistic challenges in CAD workflows, where minor topological flaws—such as missing faces or open shells—are common but difficult to detect.*** Our intention is to promote future improvements using stronger encoders, edge-aware sampling, or B-Rep-native input formats. The dataset includes metadata and ground-truth labels suitable for these enhancements.
>
> **3. Unbalanced and skewed dataset**
>
> We acknowledge the skewed storey distribution, which was an intentional design choice. Taller buildings offer greater modeling flexibility: a 10-storey sample subsumes all lower heights via truncation, while the reverse is not possible. Each building in the dataset encodes exact per-storey height (2.7 m), and it is straightforward to derive 2–9 storey variants from the 10-storey sample using B-Rep slicing tools such as OpenCASCADE. We will provide example scripts for this in the benchmark code.
>
> While our original intent was to reflect high-density urban typologies, we agree with the reviewer that balanced evaluation sets offer benefits for consistency and fairness. ***If accepted, we will release a curated balanced-storey subset (~1,500 samples per height)*** to facilitate consistent training. We also thank the reviewer for noting that storey height metadata should be made more prominent in the paper.
>
> **4. Overfitting risk due to synthetic uniformity**
>
> We agree that synthetic data may not fully reflect real-world irregularities. However, real B-Rep data is extremely scarce—precise multi-storey layouts are difficult to obtain, and floor-level interior data is often inaccessible due to privacy or ownership constraints. Even large-scale 2D datasets derived from real-world buildings, such as RPLAN (which we cite in the main text), still suffer from annotation inconsistencies—such as unreachable rooms, topological disconnections, and non-daylighted interiors—even though it is among the most carefully labeled resources in its category. These issues become harder to detect and correct in 3D, where topological validity is more structurally constrained.
>
> In contrast, BuildingBRep-11K guarantees watertight geometry and design-valid topologies by construction. While this introduces structural priors, it also yields a clean and consistent latent space—ideal for stable training, especially for geometry-aware generative models or unsupervised pretraining. Our intent is not to replace real data, but to provide a controlled and reproducible foundation for early-stage B-Rep learning. The generator (***to be released***) further allows parametric variation to explore generalization sensitivity.
>
> **5. No discussion of societal impact or safeguards**
>
> We appreciate this important reminder. We acknowledge that the current version omits a dedicated discussion of societal risks or limitations. ***If accepted, we will add a Broader Impact and Limitations section to the final version.*** While our dataset is fully synthetic and not derived from human subjects, we recognize several potential concerns: for example, the misuse of generative tools to produce unverified or unsafe building proposals, over-reliance on synthetic priors in safety-critical applications, or failure to account for local building codes.
>
> We believe these risks can be mitigated through proper framing, such as clearly marking the dataset as research-only, incorporating human-in-the-loop validation, and encouraging downstream use with domain-specific constraints. We will make these concerns and their corresponding safeguards explicit in the camera-ready version.
>
> **6. Formatting violation**
>
> We respectfully clarify that Figure 6 is ***not*** part of the 9-page main body. It belongs to the supplementary “Dataset Description” section appended after the references, which is permitted under NeurIPS guidelines. The figure is not referenced in the main text and does not affect the 9-page limit. We apologize for any confusion and will relocate it to a clearly separated appendix in the final version to prevent further misunderstanding.
>
> **Dataset code comments**
>
> We thank the reviewer for the comment on documentation. The dataset viewer on HuggingFace includes a structured table detailing all file components, and the GitHub repository provides a README with setup instructions, training/testing scripts, and dataset structure. However, we acknowledge that new users—especially those unfamiliar with 3D CAD formats—may benefit from additional guidance. ***If accepted, we will expand the documentation with step-by-step walkthroughs, loading examples, and slicing demonstrations*** to ensure broader usability across backgrounds.
>
> **Additional feedback**
>
> We appreciate the reviewer’s suggestions in the additional feedback section. As noted above, we acknowledge the simplicity of the baseline models and have clarified their intended role as learnability checks. Regarding comparisons with voxel- or mesh-based methods: while we agree that multi-format benchmarking can be valuable, our focus here is on establishing the viability of precise B-Rep input. Voxel formats, being lossy discretizations, are not fully comparable in geometric fidelity. That said, our data loader supports point-based input, and future work could explore direct mesh-based or B-Rep-native representations.
>
> We believe the concerns in Items 3–5 are addressed through our earlier responses, particularly regarding, the inclusion of a Broader Impact and Limitations section (see Point 5), justification for the dataset’s 10-storey distribution and flexibility (see Point 3), and clarification regarding formatting (see Point 6). We hope these clarifications resolve the reviewer’s concerns.

---

> > ### Comment · Reviewer_4C4u · 2025-08-05
> >
> > Thanks for the response. I have no more questions.

---

### Official Review · Reviewer_n8EJ · 2025-07-03

**Rating:** 3
**Confidence:** 3

**Summary:**

The submission introduces BuildingBRep-11K, a dataset of 11,978 procedurally generated multi-storey buildings, each represented as a precise B-rep solid with detailed per-floor metadata. Generated via a shape-grammar pipeline that enforces architectural constraints, the dataset provides clean, structured 3D models suitable for learning-based applications.

Two PointNet baselines demonstrate the dataset’s utility: one for multi-attribute regression (e.g., predicting storey count, room numbers) and another for defect classification (GOOD vs. DEFECT). Results show that the dataset is both learnable and non-trivial, supporting research in 3D building understanding and generation.

**Additional Feedback:**

No other additional feedback.

**Dataset Code Accessibility:**

No

**Dataset Code Comments:**

Although the authors have provided the code for the two experiments mentioned in the paper, the most critical component—the dataset itself—has not been released. As a result, the provided code cannot be meaningfully reproduced, which undermines its value.

**Ethical Considerations:**

No, there are no or only very minor ethics concerns

**Final Justification:**

I am still inclined to recommend the rejection of this paper.

**Limitations Weaknesses:**

1.	Storey Distribution Bias: While the authors argue that the skewed storey distribution reflects real-world building frequencies, this justification may not fully address concerns about model generalization and task adaptability. The dominance of tall buildings (e.g., 10-storey towers) could limit performance on low-rise scenarios, which are important in residential or rural contexts. Moreover, imbalanced data can lead to biased learning, and real-world distribution does not always align with the needs of benchmark design, where balanced samples often improve training stability and evaluation fairness. Since the generation is controllable, more balanced sampling strategies could have been explored.

2.	Limited Model Diversity in Benchmark Validation: The paper uses only two basic PointNet models as baselines, which limits the demonstration of the dataset’s potential. Incorporating more diverse and advanced models (e.g., PointNet++, DGCNN, Transformers) and additional tasks would strengthen the benchmark and better support future research.

3.	As a benchmark submission to NeurIPS, it is generally expected that the dataset and benchmark have established some degree of community impact or visibility. Based on the provided links and materials, the work currently appears to lack widespread attention or adoption within the research community.

**Strengths Contributions:**

This work presents a synthetic dataset of nearly 12,000 multi-storey buildings represented in boundary representation (B-rep) format, which is relatively rare in existing public datasets. Compared to prior 3D datasets like BuildingNet or City3D that rely on mesh or point cloud formats, this dataset offers more geometrically precise representations along with structured per-floor metadata, which may be useful for tasks involving spatial understanding or layout prediction.

The dataset is procedurally generated using a shape grammar that integrates basic architectural rules (e.g., room size, daylight), helping ensure consistency and plausibility. Two simple PointNet-based baselines are used to illustrate its learnability in regression and classification tasks. The paper is generally clear and well-organized, and it places the dataset in context by comparing it with both 2D (e.g., RPLAN) and 3D (e.g., Building3D) datasets. The contribution appears relevant for researchers working on 3D building generation and layout reasoning.

---

> ### Author Rebuttal · Authors · 2025-07-28
>
> **We thank the reviewer for their constructive and thoughtful feedback.**
>
> We are especially grateful for the reviewer’s recognition of our dataset’s contribution—specifically, identifying the rarity of large-scale B-Rep building data with precise geometry and structured metadata. The reviewer’s clear understanding of our goals and the value of this resource for 3D building reasoning is deeply encouraging, and it reinforces our motivation to release and support this dataset for the broader research community.
>
> We respond to each concern below.
>
> **1. Storey Distribution Bias**
>
> We acknowledge the reviewer’s concern regarding the skewed storey distribution. While this bias was an intentional design choice to support flexible modeling—since a 10-storey building subsumes all lower heights via truncation—we recognize that our justification in the paper (suggesting that it reflects real-world height frequency) was an oversimplification. In reality, storey distributions vary significantly across regions, building codes, and urban contexts. ***If accepted, we will revise this claim in the camera-ready version to reflect this complexity more accurately.***
>
> Each building in the dataset encodes exact per-storey height (2.7 m), and it is straightforward to derive 2–9 storey variants using B-Rep slicing tools such as OpenCASCADE. We will provide example scripts for this in the benchmark code.
>
> To facilitate fairer evaluation, we also plan to release a balanced-storey subset (~1,500 samples per height) that spans the full 2–10 storey range. ***We further acknowledge that the current paper does not explicate the storey height (2.7m), and we will ensure this information is properly documented and emphasized in the final version.***
>
> **2. Limited Model Diversity in Benchmark Validation**
>
> We acknowledge the reviewer’s point regarding the simplicity of our baselines. Our goal was to demonstrate the dataset’s learnability using a minimal, reproducible setup, and PointNet was chosen for its simplicity, stability, and widespread usage in geometric learning. By minimizing architectural complexity, we aimed to isolate the contribution of the dataset itself.
>
> That said, we agree that the dataset can support richer tasks and more advanced architectures. The data format is compatible with stronger 3D backbones such as PointNet++, DGCNN, and BRep-native encoders, and our codebase is modular and fully decoupled to facilitate such extensions. ***If accepted, we will include support for a second-tier benchmark using DGCNN and multi-task regression/classification to better illustrate dataset complexity.***
>
> We hope that this version of the benchmark can serve ***as a reliable foundation*** while encouraging the community to explore deeper pipelines and new task definitions based on this dataset.
>
> **3. Lack of Community Visibility or Adoption**
>
> While BuildingBRep-11K is a newly released dataset, it extends a broader line of research stemming from our prior work on ALPLAN—a 2D annotated floorplan dataset that omits doors and windows but encodes spatial layout (referenced in the main text). ALPLAN has supported multiple downstream generative studies (e.g., ***Impact of the Training Set Consistency on Architectural Plan Generation Effect Based on Pix2pixHD Algorithm***, and ***A Dual-Aspect Evaluation Framework for Architectural-Like Plan Generation via Pix2pix Series Algorithms***), which together have attracted growing attention in both machine learning and building design automation communities.
>
> BuildingBRep-11K represents the 3D geometric extension of that trajectory—grounded in similar semantic priors but encoded as watertight solids with rich parametric control. We hope it can foster structured geometric learning in CAD-like domains across ML, AEC, and automatic plan generation research.
>
> **Dataset Code Accessibility and Comments (Misunderstanding)**
>
> We respectfully clarify that the dataset is fully released, as linked in the main paper (HuggingFace). The core file is `BuildingBRep11k.tar`, which includes all 11,978 `.brep` models and the accompanying metadata in both `meta.json` and `meta.npy` formats. The benchmark code and training scripts are also available (GitHub). It is possible that the dataset links were overlooked, and we would be glad to make the access points more prominent in the final version if accepted.

---

> > ### Comment · Reviewer_n8EJ · 2025-08-08
> >
> > Thank you for the author's response. Although the author has addressed some of my concerns to a certain extent, several fundamental issues remain unresolved. Taking into account the opinions of the other reviewers as well, I am still inclined to recommend the rejection of this paper. I encourage the author to undertake substantial revisions to address the existing problems in the current version.

---

> ### Author Response · Authors · 2025-08-06
>
> Dear Reviewer n8EJ,
>
> Thank you again for your detailed and constructive review. We appreciate the clarity and specificity of your comments on storey distribution, baseline model selection, dataset visibility, and accessibility.
>
> We’ve done our best to address each of your concerns in our rebuttal—including revisions to how we frame storey statistics, plans to support more expressive baselines, and clarifications regarding dataset release and prior research contributions. If there is any part of our response that remains unclear or could benefit from follow-up discussion, we would be happy to engage further.
>
> Your feedback helped sharpen both our framing and future roadmap, and we’ve already begun integrating several of the suggestions into ongoing planning.

---

### Official Review · Reviewer_zQ4B · 2025-07-07

**Rating:** 4
**Confidence:** 4

**Summary:**

Paper propose a new 11K building brep dataset synthesis from the ground-up. Brep format for building focus more on the geometry side and is more compact than IFC. Author further demonstrate two classification tasks trained using their dataset.

**Dataset Code Accessibility:**

Yes

**Ethical Considerations:**

No, there are no or only very minor ethics concerns

**Limitations Weaknesses:**

Experiments lack compelling evidence for the use of brep format over other format. Geometry attribute prediction did not compare against a baseline with 2D top-down autocad style image. The trained network also uses point cloud as opposed to b-reps. There are multiple exisintg works like uv-net or BRepNet that directly operates on brep data for classification. It is also unclear if model trained on synthesis data would generelize to real-world data. It would be nice to have a sync-to-real experiments trained on BuildingBRep-11K.

**Strengths Contributions:**

Motivation for brep representation of building is novel. It clearly has advantage over raw IFC on certain tasks, espeically when focus is on the building geometry. In terms of writing, the paper is easy to follow and the data generation process is clearly explained. Dataset of 11k is relatively large in the building-domain and the distribution seems to be diverse for the room, floor, and storey type. Overall I think the dataset would benefit research community in the AEC domain.

---

> ### Author Rebuttal · Authors · 2025-07-28
>
> **We sincerely thank the reviewer for the thoughtful and encouraging review.**
>
> We are especially grateful for the clear recognition of our motivation for using the B-Rep format, and for the reviewer’s appreciation of the dataset’s scale, geometric fidelity, and metadata structure. We also deeply appreciate the reviewer’s recognition of its potential utility in the AEC research community. We further thank the reviewer for noting the advantages of B-Rep over raw IFC in geometry-intensive tasks, which aligns with our aim to provide a clean, directly usable solid representation. Below, we address the reviewer’s specific concerns.
>
> **1. Rationale for Choosing B-Rep over Other 3D Formats**
>
> We appreciate the reviewer’s question regarding our choice of B-Rep as the dataset format. Our decision was driven by the need for geometrically precise, information-complete, and compact 3D representations—requirements that are difficult to simultaneously meet with common alternatives.
>
> First, formats such as point clouds, meshes, and voxels were excluded early on, as they either lack exact surface definitions (point clouds), suffer from discretization artifacts (voxels), or introduce geometric approximation and topology ambiguity (meshes). These limitations make them unsuitable for tasks that require watertight solids, accurate surface boundaries, and analytic continuity.
>
> Among solid model formats, we evaluated several options including .dxf, .3ds, and .brep. We ultimately selected B-Rep because it offers:
>
> Lossless geometric representation using analytic surfaces and trimmed topology;
>
> File-level compactness—B-Rep models had consistently smaller size compared to .dxf and .3ds, even at full precision;
>
> Better compatibility with downstream geometric processing tools such as OpenCASCADE (OCP), which enables slicing, feature extraction, and solid validation.
>
> Taken together, ***B-Rep is the only format we found that satisfies all three requirements: precision, completeness, and compactness***—making it a strong fit for solid-level modeling benchmarks.
>
> We also note that learning directly from B-Rep structures is emerging as an important subfield in geometric deep learning, as exemplified by recent work like BRepNet (CVPR 2021). However, the absence of large-scale B-Rep datasets has limited progress in this area. BuildingBRep-11K aims to fill this gap and enable future research on high-fidelity solid modeling.
>
> We acknowledge that our current baselines do not directly compare B-Rep with alternative formats. However, our focus was on establishing initial learnability rather than format benchmarking. We chose B-Rep because it enables future research on tasks that require analytic surfaces and exact topology—capabilities that point clouds and meshes cannot reliably support. In future work, we plan to include direct comparisons with alternative representations and explore models that operate on B-Rep topology itself.
>
> **2. Why No 2D Plan Image Baseline Was Included**
>
> We thank the reviewer for suggesting a baseline based on 2D top-down AutoCAD-style imagery. We agree that certain geometric attributes—such as total floor area or number of rooms—can be partially inferred from rasterized floorplans, and 2D-based models have seen success in that domain.
>
> However, our benchmark tasks were designed specifically to leverage the strengths of 3D solids, including attributes that are difficult or impossible to recover from 2D alone. For example, our defect classification task relies on surface connectivity and shell completeness, which cannot be observed from top-down images. Similarly, attributes such as per-floor vertical layout, atrium presence, or tiered setback logic are inherently 3D and would not be preserved in a flat projection.
>
> That said, we recognize the value of comparing against 2D baselines to isolate the contribution of 3D input. ***If accepted, we plan to include such a baseline by rasterizing per-floor plan views from the B-Rep models and training 2D CNNs on these images.*** This will allow us to systematically compare the performance of 2D and 3D input modalities across tasks, and further clarify the conditions under which solid geometry provides a measurable advantage.
>
> **3. Use of Point Cloud Instead of B-Rep in Baselines**
>
> We thank the reviewer for pointing this out. We acknowledge that our current baselines use sampled point clouds as input rather than directly operating on B-Rep geometry. This design choice was made to ensure ***ease of reproduction and to lower the barrier*** for researchers who may not have prior experience with manifold or parametric solid representations.
>
> We fully agree that B-Rep-native models are promising tools for geometry-aware learning. Our dataset is compatible with such methods, and we plan to incorporate B-Rep benchmarks (e.g., via Euler operators or half-edge graphs) in future extensions. The current setup is intended as a ***starting point rather than a comprehensive benchmark***, and we hope it encourages broader experimentation across representation types.
>
>
> **4. Lack of BRep-Specific Baselines (e.g., BRepNet, UV-Net)**
>
> We acknowledge the reviewer’s suggestion to use BRepNet and UV-Net, both of which represent important recent developments in B-Rep-specific deep learning. At the time of our implementation, we focused on verifying dataset learnability using simple and widely adopted baselines. To our knowledge, BRepNet has only published support for surface segmentation tasks; it was unclear whether it could be directly adapted for binary classification without significant architectural modifications.
>
> UV-Net, while supporting classification, is designed for watertight CAD models with surface parameterizations. Our second task—detecting whether a B-Rep model is topologically defective—requires handling open-shell, incomplete solids with missing faces, for which UV mapping may not be well-defined. As such, it is unclear whether UV-Net can be directly applied to this task without significant adaptation.
>
> Both models offer valuable directions for future research, and we hope that ***BuildingBRep-11K can serve as a platform for extending such architectures*** to handle noisy or incomplete solids more robustly.
>
> We believe our current classification task fills a practical and underexplored need—detecting topological defects in solid models—and highlights a limitation in current B-Rep architectures that we hope future work will address.
>
> **5. Synthetic-to-Real Generalization**
>
> We acknowledge the reviewer’s concern about generalization to real-world data. We agree that ***models trained solely on synthetic data cannot be directly deployed without further validation on real-world B-Rep buildings***. Unfortunately, such datasets are currently extremely scarce—both due to the high cost of creating watertight B-Rep models at architectural scale, and because of privacy and proprietary constraints in releasing building-level design data.
> Even in the 2D domain, widely used datasets like RPLAN (which we reference) suffer from annotation inconsistencies such as unreachable rooms or disconnected spaces. These challenges are only magnified in 3D, where modeling clean solid geometry requires exact surface definitions, tight topological closure, and component-level annotation—all of which are difficult to extract or verify at scale. Nonetheless, ***we believe such real-world B-Rep datasets will eventually emerge. Our goal is to ensure that, when they do, the community will already have a foundation of pretrained models, benchmarks, and learning pipelines ready to build upon***.
> To mitigate potential bias and support broader research needs, ***we will release the full procedural generator*** along with the dataset. It exposes over 30 adjustable parameters—including storey count, spatial layout, massing logic, and vertical core design—enabling users to synthesize customized samples with different geometric or spatial characteristics. This flexibility is difficult to achieve in real-world pipelines, especially across thousands of buildings.
>
> While synthetic data may not fully reflect real-world irregularities, its consistency and controllability make it ***a powerful incubator*** for developing transferable B-Rep-specific learning methods.
>
> **Closing Note**
>
> We thank the reviewer for their insightful and technically grounded feedback. The suggestions regarding baseline diversity, representation comparisons, and real-world generalization have helped us sharpen both the scope and future direction of our work. We acknowledge the current limitations in our benchmark design and implementation, and we are committed to extending the dataset’s utility through stronger baselines, broader input modalities, and more flexible generation tools. We hope BuildingBRep-11K can serve as a stepping stone for the community to explore high-fidelity solid modeling at scale.

---

> ### Author Response · Authors · 2025-08-06
>
> Dear Reviewer zQ4B,
>
> Thank you again for your thoughtful and technically grounded review. Your detailed feedback on our design choices, baselines, and real-world generalization has been extremely helpful.
>
> We’ve done our best to address each of your points in our rebuttal—including the rationale for using B-Rep, the exclusion of 2D image baselines, the current use of point clouds, and the limitations of our model selection. If any aspects of our response remain unclear, we would be very happy to discuss them further.
>
> Your comments helped us sharpen both the scope and future direction of this work, and we’ve already begun incorporating some of the suggestions into our ongoing planning.

---

### Official Review · Reviewer_RMDh · 2025-07-22

**Rating:** 3
**Confidence:** 1

**Summary:**

This submission introduces BuildingBRep-11K, the first large-scale dataset of 11,978 multi-storey buildings represented in precise B-Rep (boundary representation) format, complete with rich layout metadata. The buildings are synthetically generated using a shape-grammar-driven pipeline that incorporates realistics architectural rules such as daylighting, interior circulation, and spatial hierarchy. Each instance includes a watertight 3D B-Rep solid and a .npy metadata file specifying per-floor and per-room attributes.

To demonstrate the dataset’s utility, the authors provide two baseline experiments using a lightweight PointNet architecture:

1. Multi-attribute regression, predicting storey count, total rooms, per-storey room counts, and mean room area from a 4k-point cloud.

2. Defect detection, classifying models as GOOD or DEFECT (with missing geometry), showing promising recall despite limited precision.

**Dataset Code Accessibility:**

Yes

**Dataset Code Comments:**

The dataset and code are publicly available through HuggingFace and GitHub, as linked in the paper.

**Ethical Comments:**

No, there are no or only very minor ethics concerns

**Ethical Considerations:**

No, there are no or only very minor ethics concerns

**Limitations Weaknesses:**

1. The two learning baselines use a simple PointNet architecture without benchmarking against stronger or more recent 3D models (e.g., DGCNN, PointTransformerV3,).
2. While the dataset is clearly valuable for research and development, the paper could benefit from a brief reflection on potential limitations or considerations when applying synthetic architectural data in real-world contexts. For instance, acknowledging differences between synthetic and real-world design constraints, or possible risks in over-relying on procedural data for downstream applications, would strengthen the paper's completeness and transparency.
3. As noted in the defect classification task, the training split contains a 2:1 ratio of defective to good models, which may skew learning. This imbalance leads to high recall but low precision. Although acknowledged by the authors, the classifier performance could be further improved by experimenting with class-balancing techniques, such as weighted loss functions or oversampling.

**Strengths Contributions:**

1. Introduces a large-scale collection (11,978 samples) of multi-storey buildings in precise, watertight B-Rep form—filling a clear gap in existing 3D building resources that are mostly mesh or point-cloud based.

2. Data are generated through a rule-driven pipeline grounded in architectural practice (scale controls, daylight access, circulation, openings, tiered stacking), producing diverse yet plausible building forms.

3. Each model ships with structured per-floor/room metadata, and the full dataset plus generation and benchmark code are publicly released for reuse and extension.

4. Baseline experiments (multi-attribute prediction; good/defect classification) show the dataset is tractable yet non-trivial for learning-based methods, underscoring its value as a benchmark.

---

> ### Author Rebuttal · Authors · 2025-07-29
>
> **We sincerely thank the reviewer for their thoughtful and constructive review.**
>
> We deeply appreciate the recognition of BuildingBRep-11K’s contribution as a large-scale, watertight B-Rep dataset with structured per-floor and per-room metadata—filling a gap in current 3D building resources. We are also grateful for the reviewer’s careful attention to both the design of our generation pipeline and our implementation of the benchmark tasks. Below, we address each of the reviewer’s concerns in turn.
>
> **1. Limited Baseline Model Diversity**
>
> We acknowledge that the two baselines in our current version rely on a basic PointNet architecture. This was a deliberate choice: as our background lies primarily in building computing rather than model architecture design, we ***prioritized simplicity and accessibility***. Our intention was to verify dataset learnability using a widely adopted and reproducible backbone.
>
> That said, we fully agree that the dataset is well-suited for stronger and more diverse 3D encoders, including DGCNN, PointTransformer, and BRep-native models. The codebase is modular and decouples data loading from model definition, enabling easy integration of such architectures. We welcome community contributions in this direction and, ***if accepted, plan to include a second-tier benchmark using DGCNN in a future update***.
>
> **2. Lack of Discussion on Synthetic-to-Real Transfer Considerations**
>
> We appreciate this important point. We acknowledge that the current version does not explicitly reflect on the limitations of using synthetic data for real-world applications. If accepted, we will ***add a Broader Impact and Limitations section*** clarifying that BuildingBRep-11K is a fully synthetic dataset and ***outlining potential risks***—such as over-reliance on procedural priors or failure to capture region-specific design constraints—when deploying models trained solely on synthetic data.
>
> We would also like to elaborate on why we believe synthetic data still holds value in this context. Real-world multi-storey precise 3D solid data is extremely difficult to acquire, both due to privacy concerns and the challenges of modeling clean solid geometry. Even in 2D, widely used datasets like RPLAN (which we reference) suffer from annotation errors such as unreachable rooms, disconnected spaces, or lightless interiors. These issues become even more pronounced in 3D.
>
> By contrast, our dataset guarantees watertight geometry and topological validity by construction. While this introduces regularity, it also enables stable training and controlled benchmarking. We hope that BuildingBRep-11K can ***serve as an incubator for developing algorithms and methods for precise 3D solid modeling***—so that when real, high-quality B-Rep data becomes available in the future, the community will already have robust models and workflows ready to build upon.
>
> As a remedy to the fixed nature of the current procedural rule, and to support diverse downstream needs, ***the generator will be released along with the dataset if accepted***. It exposes approximately 30 modifiable parameters—including storey count, unit depth, facade layout, and vertical core topology—allowing users to generate customized variants under their own spatial or design constraints. ***This flexibility is difficult to achieve in real-world pipelines, especially at scale, and we believe it will significantly broaden the dataset's utility***.
>
> **3. Class Imbalance in Defect Classification**
>
> We thank the reviewer for highlighting this important point. We agree that the 2:1 imbalance in the GOOD/DEFECT split may affect classifier behavior—particularly in terms of limited precision despite high recall. As noted in the paper, this skew reflects the intrinsic difficulty of generating flawless B-Rep models under procedural variation, where minor topology errors are frequent and often subtle.
>
> We see the defect classification task as a realistic and meaningful challenge, especially for benchmarking model robustness under noisy or imbalanced inputs. That said, we agree that model performance can be improved with class-balancing techniques. ***If accepted, we will include updated baselines using weighted loss functions and oversampling strategies,*** and report full precision, recall, and F1 metrics to provide a more complete evaluation.
>
> **Closing Note**
>
> ***We appreciate the reviewer’s thoughtful comments and suggestions, which have helped us reflect critically on both the strengths and limitations of our work.*** In addition to the specific revisions described above, we are committed to improving the dataset's accessibility, flexibility, and alignment with real-world needs. We believe BuildingBRep-11K offers a strong foundation for research on precise 3D solid modeling, and we hope it will serve the community not only as a training resource, but also as a stepping stone toward broader benchmark design and application-aware modeling tasks.

---

> ### Author Response · Authors · 2025-08-06
>
> Dear Reviewer RMDh,
>
> Thank you again for your thoughtful and detailed review of our submission. We sincerely appreciate the time you took to assess our work and the constructive feedback you provided.
>
> We’ve done our best to address your concerns in our rebuttal—including the discussion on baseline model diversity, the limitations of synthetic data for real-world generalization, and the effect of class imbalance on the defect classification task. If there are any parts of our response that remain unclear or warrant further clarification, we would be very glad to engage further.
>
> Your suggestions offered valuable insights that we have already begun incorporating into our ongoing research efforts.

---

### Note · Authors · 2025-08-14

Thank you to the reviewers and AC for the constructive evaluation and discussion. We briefly summarize clarifications and our commitments.

**Clarifications**

• The dataset, metadata, and benchmark code were publicly released at submission time; the paper references the access points.

• Figure 6 appears after the references in a supplementary “Dataset Description” section, is not counted toward the 9-page main body, and is not referenced in the main text.

• Baselines are intentionally lightweight to verify learnability and maximize reproducibility; the data are fully compatible with stronger 3D backbones and B-Rep-native models.

• The skew toward 10-storey buildings is intentional to maximize vertical complexity; each sample encodes a 2.7 m storey height, so lower-storey subsets can be obtained by simple slicing.

**Commitments if accepted**

• Release the procedural generator (≈30 parameters) with clearer docs and examples, including slicing scripts and usage guides.

• Provide a balanced 2–10-storey subset and update the defect task with class-weighted/oversampling baselines; report full precision/recall/F1.

• Add a Broader Impact & Limitations section describing synthetic-to-real risks and recommended safeguards.

• Extend the benchmark with a DGCNN baseline and B-Rep-native hooks to ease future model integration.

**Why acceptance matters**

BuildingBRep-11K fills a clear gap: a large-scale, watertight B-Rep corpus with rich per-floor metadata, enabling tasks that require exact geometry/topology (volumetric reasoning, vertical slicing, section reconstruction). We believe this resource can catalyze B-Rep learning and generation methods so that, when real-world B-Rep corpora become available, the community already has robust tasks, models, and pipelines to build upon.

We appreciate your consideration.

---

### Decision · Program_Chairs · 2025-09-18

**Decision:**

Reject

**Comment:**

The paper contributes a benchmark of multi-storey buildings representation in B-Rep (boundary representation) format.  This data supports research on methods to analyze and generate 3D representations of buildings.  The work appears to be technically sound and on a topic of relevance to NeurIPS.

The initial reviews identified some weaknesses and concerns, with key issues about: missing stronger baseline methods beyond PointNet architecture, lack of discussion of limitations of synthetic data relative to real-world data, lack of evidence about the benefits of B-Rep vs alternative 3D representations, and biases of the dataset due to distribution over building sizes in terms of the number of stories.

After the rebuttal and discussion, two reviewers found their concerns were not sufficiently addressed and thus remained negative, one initially negative reviewer did not update their opinion, and lastly one initially positive reviewer also did not participate in discussion or update their opinion based on other reviews and the discussion. Due to several remaining concerns as described above, the AC does not find a basis to overrule the majority reviewer opinion which recommends rejection.  It is recommended to incorporate improvements and suggestions from this review cycle to strengthen the work and target an upcoming venue for publication.